# Modulation of MicroRNA Processing by Dicer via Its Associated dsRNA Binding Proteins

**DOI:** 10.3390/ncrna7030057

**Published:** 2021-09-16

**Authors:** Toyotaka Yoshida, Yoshimasa Asano, Kumiko Ui-Tei

**Affiliations:** Department of Biological Sciences, Graduate School of Science, The University of Tokyo, Tokyo 113-0033, Japan; toyotaka-yoshida104@g.ecc.u-tokyo.ac.jp (T.Y.); yoshimasa.asano@bs.s.u-tokyo.ac.jp (Y.A.)

**Keywords:** microRNA biogenesis, Dicer-associated proteins, dsRBP, TRBP, ADAR, PACT, LGP2, miRNA–mRNA network

## Abstract

MicroRNAs (miRNAs) are small non-coding RNAs that are about 22 nucleotides in length. They regulate gene expression post-transcriptionally by guiding the effector protein Argonaute to its target mRNA in a sequence-dependent manner, causing the translational repression and destabilization of the target mRNAs. Both Drosha and Dicer, members of the RNase III family proteins, are essential components in the canonical miRNA biogenesis pathway. miRNA is transcribed into primary-miRNA (pri-miRNA) from genomic DNA. Drosha then cleaves the flanking regions of pri-miRNA into precursor-miRNA (pre-miRNA), while Dicer cleaves the loop region of the pre-miRNA to form a miRNA duplex. Although the role of Drosha and Dicer in miRNA maturation is well known, the modulation processes that are important for regulating the downstream gene network are not fully understood. In this review, we summarized and discussed current reports on miRNA biogenesis caused by Drosha and Dicer. We also discussed the modulation mechanisms regulated by double-stranded RNA binding proteins (dsRBPs) and the function and substrate specificity of dsRBPs, including the TAR RNA binding protein (TRBP) and the adenosine deaminase acting on RNA (ADAR).

## 1. Introduction

MicroRNAs (miRNAs) are single-stranded RNAs of approximately 22 nucleotides in length and are classified as small non-coding RNAs. The miRNAs regulate gene expression post-transcriptionally by a mechanism known as RNA silencing, where miRNA is loaded onto Argonaute (AGO), a core component of the miRNA-induced silencing complex (miRISC) [1]. While on AGO, the miRNA recognizes target mRNAs that have sequences that are complementary to the “seed region” (positions 2–8 from the 5′ end) of the miRNA in their 3′ untranslated region [2]. As the seed region only consists of seven nucleotides, each miRNA is capable of recognizing and regulating many types of mRNAs, indicating that miRNA–mRNA gene expression networks are highly complicated.

Some of the miRNAs that have been discovered in diverse eukaryotes are evolutionally conserved, while others are species specific [3,4,5,6]. The first two miRNAs, lin-4 and let-7, were discovered in *Caenorhabditis elegans* (*C. elegans*) through the analysis of the heterochronic gene mutants that undergo development/differentiation at an abnormal time within the organism [7,8,9]. Let-7 is evolutionally conserved across various species, including in humans. In human lung cancer cells, let-7 regulates cell proliferation and also suppresses the expression levels of NRAS and KRAS, two genes that induce oncogenic transformation when mutated [10,11,12]. Thus, although let-7 is conserved in both *C. elegans* and humans, its function in the two species is different. As for species-specific miRNAs, more than 2000 human and mouse miRNAs have been registered in the miRBase database [13], whereas about 400 miRNAs have been registered for *Drosophila melanogaster* and *C. elegans* as well as about 200 miRNAs for *Xenopus*, suggesting that higher organisms may have more miRNAs compared to lower organisms. The species-specific miRNAs in the higher organisms play an important role in divergent physiological phenomena, including development, differentiation, apoptosis, and cell growth [14]. Human-specific miRNAs are often associated with cognition and neurological disorders [15].

At the initial stage, miRNA is transcribed by RNA polymerase II (Pol II) as primary-miRNA (pri-miRNA), which has stem loop structures [16,17] (Figure 1). In the canonical miRNA biogenesis pathway, the flanking regions of the pri-miRNA are cleaved to generate precursor-miRNA (pre-miRNA) in the nucleus by a microprocessor complex consisting of Drosha, a member of the RNase III family proteins, and its cofactor DiGeorge syndrome critical region 8 (DGCR8), a double-stranded RNA (dsRNA) binding protein (dsRBP) [18,19,20]. The pre-miRNA is transported from the nucleus to the cytoplasm by Exportin-5 (EXP5), which then couples with GTP-bound Ran [21]. In the cytoplasm, Dicer, an RNase III family protein, cleaves off the loop region of the pre-miRNA to generate a miRNA duplex in collaboration with the trans-activation response (TAR) RNA binding protein (TRBP) in the canonical miRNA biogenesis pathway [22,23]. The interaction of the Dicer–TRBP complex with Argonaute (AGO) facilitates the loading of the miRNA duplex onto AGO to form the RISC-loading complex (RLC) [24,25,26]. The miRNA duplex is then unwound into single stranded miRNAs; the RNA strand that remains on the AGO protein acts as the miRNA, while the other strand is discarded [27]. The former is called the guide strand, and the latter is called the passenger strand. The mature miRNA on the AGO protein guides the RISC to target mRNAs that have sequences that are complementary to the seed region of the miRNA. After binding to the mRNA, AGO recruits the trinucleotide repeat containing 6 (TNRC6) protein, a scaffold protein tethering effector proteins to destabilize and translationally repress target mRNAs by inducing their decapping and deadenylation [28].

In mice, the knockout of *Drosha* and *Dicer* resulted in global miRNA deficiencies, which suggests that both proteins are required for miRNA biogenesis [29,30,31,32,33,34]. While both of them are classified as RNase III family proteins, their recognition mechanisms for substrates and cofactors differ greatly [35,36]. Processing by Drosha requires DGCR8 as a dedicated partner, unlike processing by Dicer, which does not require other proteins. However, some Dicer-associated proteins are involved in the enhancement/inhibition of Dicer activity. Here, we introduce the processing machinery of Drosha and Dicer and the regulated substrate selectivity of Dicer.

## 2. Processing of Pri-miRNA by Drosha-DGCR8 Microprocessor

In the nucleus, pri-miRNA is transcribed from the genome as a single stranded RNA that is hundreds to thousands of nucleotides in length and is cleaved by the microprocessor complex, which comprises one Drosha molecule and two DGCR8 molecules [37] (Figure 2a,b). The pri-miRNA forms a stem–loop hairpin structures with flanking regions [38,39]. The terminal loop of pri-miRNA is recognized by DGCR8, and the stem–flank junction of the pri-miRNA is recognized by the central domain (CED) of Drosha [40,41,42,43,44] (Figure 2b). The microprocessor measures the nucleotide length from the stem–flank junction of the pre-miRNA, and cleavage is induced by the intramolecularly dimerized RNase III domain a (RIIIDa) and RIIIDb (Figure 2a). As a result, a pre-miRNA with a 2-nucleotide overhang at the 3′ end is generated [45].

Both Drosha and DGCR8 are classified as dsRBP superfamily proteins, which have dsRBDs consisting of 65–70 amino acids [46,47,48] (Figure 2a). The dsRBDs of Drosha and DGCR8 contribute to pri-miRNA binding (Figure 2b). The dsRBD binds to dsRNA via a highly conserved αβββα motif [49,50]. Structural analysis indicates that dsRNA is recognized by three regions of the αβββα motif: the N-terminal α-helix 1 (α-1) (region 1); the loop region between β-strand 1 (β-1) and β-strand 2 (β-2) (region 2), which interact with different minor grooves of the dsRNA; and the C-terminal α-helix 2 (α-2) (region 3), which interacts with a major groove between the two minor grooves [51,52] (Figure 3). Regions 1 and 2 mainly interact with the 2′-OH group of ribose in the minor groove of RNA, which enables the dsRBD to distinguish RNA from DNA. Region 3 interacts with the phosphodiester backbone of both strands of the major groove, which allows dsRBD to specifically recognize the A-form helical structure with a narrow and deep major groove rather than the B-form helical structure with a wide and shallow major groove. Unlike other dsRBDs, an additional six amino acids are inserted in the linker region between the α-1 and β-1 in dsRBD of Drosha, and their loss decreases its cleavage activity [53]. In contrast, the insertion of these amino acids into the linker region in the dsRBDs of other dsRBPs decreases their ability to bind to dsRNAs, suggesting that dsRBDs have intricate structures and functions according to the perturbations in the linker region. Recent reports have also indicated that the dsRBD of Drosha structurally recognizes the GHG motif (in which H is any nucleotide except G) conserved in the stem regions of some pri-miRNAs [54,55] (Figure 2b). Several other sequence motifs are conserved in pri-miRNAs, as shown in Figure 2b, and they facilitate the correct recruitment of microprocessors to pri-miRNAs [37,56,57]. The microprocessor cleaves pri-miRNAs the through recognition of the structural features of pri-miRNA by Drosha and DGCR8. It has been reported that the embryonic stem cells from *DGCR8*-knockout mice showed global miRNA deficiency, which indicates that DGCR8 is essential for the biogenesis of miRNAs [58]. To date, no other Drosha partner proteins are required for the processing of pri-miRNA have been identified, suggesting that DGCR8 may be Drosha’s only essential partner for pri-miRNA cleavage. However, in unusual cases, the enhancer of rudimentary homolog (ERH) was reported as an additional component of the microprocessor, and it facilitates the efficient processing of suboptimal hairpin pri-miRNAs, including the miR-451 hairpin, through the induction of the microprocessor multimerization [59].

## 3. Processing of Pre-miRNA by Dicer

In eukaryotes, the domain structures of Dicer proteins are widely conserved; however, depending on the species, there are different mechanisms of substrate recognition. *Drosophila* has two Dicer paralogs that have different functions: Dicer-1 (Dcr-1) acts on the miRNA maturation pathway, and Dcr-2 acts on the small interfering RNA (siRNA) pathway, the purpose of which is for dicing the long dsRNA [60]. Plants also have at least four distinct classes of Dicer-like (DCL) proteins (DCL1–4) [61]. The DCL1 functions for miRNA processing and the DCL2-4 redundantly function for siRNA production. Humans, on the other hand, have only one *Dicer* gene, which functions in both pathways. It has been reported that Dicer cleaves pre-miRNAs more efficiently than dsRNAs for siRNA production in vitro [62,63,64,65]. Human Dicer is a multiple-domain protein (Figure 2a). An electron microscopy study demonstrated that Dicer forms an L-shaped structure [66,67] and recognizes the 3′-overhang of pre-miRNA by the Piwi, Argonaute, and Zwille (PAZ) domain, while the phosphorylated 5′-end of the pre-miRNA is captured by the platform domain [68,69] (Figure 2c). These two domains are arranged to be able to recognize the structure of pre-miRNA [70,71]. In the 5′ and 3′ counting rules, Dicer measures the nucleotide lengths from both ends of the pre-miRNA and cleaves the terminal loop of the pre-miRNA by means of the intramolecular dimerization of two RNase III domains, RIIIDa and RIIIDb, generating a miRNA duplex [45,72]. Unlike human Dicer, *Drosophila* Dcr-1 recognizes the terminal loop of pre-miRNA by its DExD/H-box helicase domain and specifically cleaves pre-miRNAs in a loop size-dependent manner [73]. On the other hand, *Drosophila* Dcr-2 recognizes the blunt end of dsRNA via its helicase domain and processes the dsRNA for siRNA production [74]. The DExD/H-box helicase domain of human Dicer does not have such substrate selectivity. The dsRBD of human Dicer also cannot distinguish the stem region of the pre-miRNA from that of the dsRNA of the siRNA in vitro [75], indicating that Dicer, on its own, has low substrate selectivity.

## 4. Enhancement of Pre-miRNA Processing by Dicer via TRBP/PACT

### 4.1. The Role of TRBP and PACT in Pre-miRNA Processing by Dicer

Dicer-associated proteins regulate the substrate recruitment and the cleavage activity of Dicer [76,77]. The processing of pre-miRNA is not only promoted by TRBP but also by other Dicer-associated proteins, such as PACT, a protein activator of protein kinase R (PKR) [76,78,79]. TRBP is a protein that binds to TAR RNA, a hairpin-structured RNA that is encoded by human immunodeficiency virus type I [80]. PACT is a protein that was initially identified as an activator of PKR [81]. TRBP and PACT have highly conserved domain structures with three dsRBDs [82] (Figure 2a). The dsRBDs are divided into two subclasses, type-A and type-B. Type-A has a conserved αβββα motif with a high affinity to dsRNA [49,50]. Type-B, also termed half dsRBD, has poorly conserved N-terminal sequences in the αβββα motif and is associated with protein–protein interactions [83,84]. The first and second dsRBDs of TRBP and PACT are type-As, and the third dsRBD is a type-B that binds to Dicer through the DExD/H-box helicase domain [76,78,79].

Both TRBP and PACT interact with Dicer to promote the cleavage of the pre-miRNA in the RLC containing AGO protein and the facilitate loading of the miRNA duplex onto AGO [23,26,76,85,86]. It was reported that deletion or mutation of the DExD/H-box helicase domain of Dicer activated the cleavage of its substrates, which suggested that this domain inhibits catalytic activity rather than affecting RNA-substrate binding [87]. TRBP binds to the DExD/H-box helicase domain of Dicer and stimulates the cleavage activity of Dicer. Therefore, the DExD/H-box helicase domain functions as an intramolecular structural switch that maintains Dicer in a low-activity state until the partner proteins interact with its DExD/H-box helicase domain. In addition, it was reported that TRBP facilitates the processing activity of pre-miRNA by Dicer in RNA-crowded molecular environments [88] and that it also facilitates the recruitment of pre-miRNAs to the PAZ domain of Dicer. Furthermore, the sliding motion of TRBP on dsRNA with Dicer has been reported [89]. This was associated with the higher substrate cleavage activity of Dicer compared to Dicer alone, which suggests that TRBP facilitates the cleavage activity of Dicer by guiding Dicer to the substrates. To date, no studies on the mechanism by which PACT promotes Dicer-mediated cleavage of pre-miRNAs have been reported. However, the amino acid sequence of the Dicer-interacting dsRBD of PACT was found to be similar to that of TRBP. It has yet to be determined if PACT interacts with Dicer by a mechanism similar to that of TRBP and if it enhances the processing of similar types of pre-miRNAs.

TRBP and PACT have different functions. Although TRBP preferentially binds to simple duplex RNA, PACT inhibits Dicer-mediated dsRNA cleavage for siRNA production [90]. Unlike PACT, the cleavage site for Dicer-TRBP shifts when compared to cleavage by Dicer alone [91,92]. PACT and TRBP have no redundant effects on the production of isomiRs, different-sized miRNAs that alter the downstream target-binding specificities. Such differences in dsRNA recognition and processing behavior are attributed to two N-terminal RNA-binding domains in each protein.

### 4.2. TRBP-Mediated Regulation of Specific miRNA Maturation by Dicer

The maturation of miRNA is promoted by TRBP binding. Using RNA immunoprecipitation sequencing (RIP-seq), we recently demonstrated that the secondary structures of pre-miRNAs may differ in TRBP-bound and non-bound pre-miRNAs [93] (Figure 4). In this analysis, we used the base-pairing probability (BPP) of miRNAs. The BPP score represents the probability of base-pairing with respect to an ensemble of RNA secondary structures that are available for the prediction of accurate RNA secondary structures [94,95]. The BPP values for the stem regions of TRBP-bound pre-miRNAs were higher than the mean values of all of the pre-miRNAs, with the exception of the central regions. TRBP-non-bound pre-miRNAs exhibited low BPP values. These results indicate that TRBP preferentially binds to pre-miRNA with tight base-pairing. It was also reported that the dsRBDs of TRBP bind to siRNA in a sequence-independent manner [96]. These results suggest that the structural features of pre-miRNAs, including mismatches and bulges, are important for TRBP-dependent substrate recognition.

Several studies have addressed the TRBP-mediated maturation of specific miRNAs and its effect on downstream pathways. It was reported that the TRBP-mediated maturation of miR-208a decreased the expression level of SRY-Box Transcription Factor 6 (Sox6), which is required for normal heart function [98]. It was also reported that disruptions of TRBP-dependent maturations of tumor suppressor certain miRNAs (TS-miRs), miR-143 and miR-145, were related to the self-renewal and tumor maintenance of cancer stem cells [99]. These results suggest that TRBP regulates biogenesis and the downstream gene regulatory pathways of specific miRNAs.

## 5. Enhancement of Pre-miRNA Processing by Dicer via ADAR1

### 5.1. ADAR1-Mediated Promotion of Pre-miRNA Processing by Dicer

ADAR1 is classified as an adenosine deaminase acting on the RNA (ADAR) family protein that edits adenosine into inosine on dsRNA (A-to-I RNA editing) [100,101,102]. In addition to having zinc finger domains, ADAR1 has three dsRBDs and a deaminase domain (Figure 2a). The ADAR1-mediated regulation of miRNA biogenesis is classified into two types: one is the regulation of Drosha and Dicer cleavage by the A-to-I RNA editing of their substrate miRNAs/siRNAs [103,104,105], and the other is the promotion of miRNA maturation, which is achieved by forming a complex with Dicer via protein–protein interaction [85]. ADAR1 interacts directly with the DExD/H-box helicase domain of Dicer via its second dsRBD in the absence of dsRNA [85], while all ADAR dsRBDs that are type-A are involved in binding to dsRNA. Thus, the interaction mechanism between ADAR1 and Dicer is different from that of TRBP–Dicer or PACT–Dicer interaction. It has been reported that the DExD/H-box helicase domain of Dicer is essential for interaction with dsRBP during viral infection [106]. ADAR1 facilitates Dicer-mediated pre-miRNA cleavage and loading onto RISC for miRNA maturation and siRNA production [85]. However, the detailed mechanism for promoting Dicer-mediated processing by ADAR1 is not clear. Unlike TRBP and PACT, ADAR1 may regulate pre-miRNA processing by Dicer in combination with additional deaminase domain activity.

### 5.2. ADAR1-Mediated Maturation of Specific miRNAs

In vertebrates, there are three types of ADAR proteins, two of which have catalytic activities (ADAR1 and ADAR2). ADAR1 also has two splice variants; N-terminally truncated ADAR1p110 is mainly localized in the nucleus, and the full-length ADAR1p150 is induced by interferons in the cytoplasm [107,108]. Recently, we reported that ADAR isoforms may bind to specific pre-miRNAs depending on the secondary structures predicted by BBP in the predominantly double-stranded region of each pre-miRNA [97] (Figure 4). ADAR1p110-bound pre-miRNAs had higher BPP values in their stem regions near the 3′-overhang side compared to the control. ADAR2-bound pre-miRNAs showed higher BPP values in their central region and the 3′-overhang side. These results suggested that ADAR1p110 binds more effectively to pre-miRNA with incomplete base-pairing near the center of the stem region than ADAR2 does. Thus, ADARs have substrate selectivity based on the secondary structure of the dsRNA.

ADAR1 regulates the maturation of specific miRNAs in a spatiotemporal manner by interacting with Dicer. It was reported that *Adar*-knockout mice underwent systemic apoptosis at embryonic day 12 (E12) followed by death [109]. Analysis of these *Adar1*-knockout mice showed that the expression of ADAR1 and Dicer increased gradually from E9 to E12, and the expression of specific miRNAs, including miR-1 and miR-181a, increased at E12, suggesting the importance of miRNA maturation via Dicer–ADAR1 interaction during embryogenesis [85]. Several research groups have reported that miRNA maturation mediated by Dicer-ADAR1 affects global gene expression profiles in various diseases. It was reported that ADAR1p150 promotes the maturation of specific miRNAs by interacting with Dicer during viral infection [110]. In mice models, the overexpression of ADAR1p150 induced the expression of miR-222, which, in turn, repressed the expression of phosphatase and the tensin homolog (PTEN), an apoptosis-related gene, leading to increased cell survival. A recent report indicated that the upregulation of the ADAR1 expression level facilitated the Dicer-mediated processing of specific miRNA in oral cancer patients [111]. The expression levels of six oncogenic miRNAs (onco-miRs) were increased by Dicer–ADAR1 interaction. These results suggested that ADAR1-mediated miRNA maturation regulates the downstream gene pathways.

## 6. TRBP-LGP2 Interaction Inhibits Pre-miRNA Processing by Dicer

In virus infected mammalian cells, virus-derived RNAs are captured by viral sensor proteins such as retinoic acid-inducible gene I-like receptors (RLRs), inducing the production of type I interferon [112,113,114]. Recently, we reported that the TRBP-mediated maturations of pre-miRNAs were inhibited through the competitive binding of the laboratory of genetics and physiology 2 (LGP2) to Dicer–TRBP interaction during Sendai virus infection [115]. Interferons enhanced the expression of LGP2, which interacted with TRBP to inhibit the Dicer–TRBP interaction [93]. Following LGP2-dependent inhibition of Dicer–TRBP interaction, the maturations of TRBP-bound pre-miRNAs, including miR-106b, were suppressed. The inhibition of the maturation of such miRNAs increased the expression of apoptosis-related genes downstream of miRNA processing. This finding suggested that the crosstalk between antiviral response and miRNA biogenesis is regulated by TRBP binding to the specific pre-miRNAs.

## 7. Concluding Remarks

Research on miRNAs accelerated significantly in the early 2000s, and since then, tens of thousands of studies have been reported annually. MiRNAs are widely conserved in eukaryotes and stand as one of the key regulators of gene expression networks. In this review, we summarized and discussed recent reports addressing the modulation of the substrate selectivity of Dicer by its associated proteins during miRNA biogenesis. Unlike Drosha, Dicer-mediated processing of pre-miRNA is controlled by multiple Dicer-associated proteins through the recruitment of pre-miRNAs to Dicer and the regulation of Dicer cleavage activity. We reported that the prediction of pre-miRNA secondary structures based on BPP could be used as an indicator to predict the substrate selectivity of Dicer-associated proteins because those proteins, which regulate Dicer-mediated processing, are usually dsRBPs [93,97]. Thus, the species of miRNAs that undergo processing by Dicer could be different depending on the specific Dicer-associated proteins, such as TRBP and ADAR1, and this may explain why the gene expression network in downstream pathways largely differs (Figure 5). However, there are still many challenges regarding the significance of such complicated gene expression networks that are regulated by miRNAs. The first challenge is to identify additional Dicer-associated proteins, which can regulate miRNA biosynthesis. The second challenge is to establish a highly accurate RNA secondary structure prediction program, as miRNAs are generally structurally complicated, including components such as mismatches, wobble base pair, or bulge structures. Alternatively, biochemical experiments such as selective 2′-hydroxyl acylation analyzed by primer extension sequencing (SHAPE-seq) and structure-seq may be also useful to predict the structure of miRNAs with a resolution of one nucleotide [116,117].

Studies have reported that there are close connections between miRNAs and diseases and that there are possibilities for miRNAS to be used as biomarkers [11,118]. The development of antisense oligonucleotides targeting disease-causing miRNAs is underway [119]. A precise understanding of the miRNA-mediated gene regulatory mechanisms is also necessary for miRNA nucleic acid therapeutics.

## Figures and Tables

**Figure 1 ncrna-07-00057-f001:**
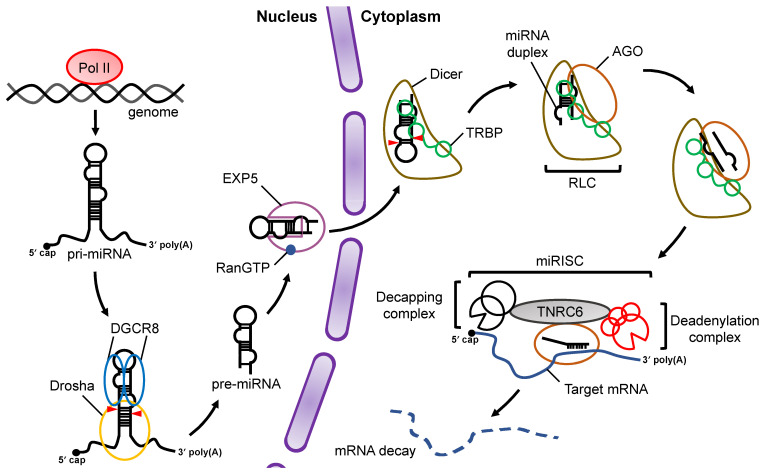
Overview of miRNA biogenesis and the RNA silencing pathway. Pri-miRNAs are transcribed from the genome by Pol II. In the nucleus, pri-miRNA is cleaved by a microprocessor complex consisting of Drosha and DGCR8 to produce pre-miRNA, which is then transported from the nucleus to the cytoplasm by EXP5 coupled with GTP-bound Ran (RanGTP). Cleavage of pre-miRNA is performed by Dicer and its cofactor, TRBP, in a canonical miRNA biogenesis pathway. After pre-miRNA cleavage, the miRNA duplex is loaded onto AGO proteins through formation of the RLC complex. The mature miRNA guides the RISC complex to target mRNAs that are complementary to the seed region of miRNA. Recruitment of the TNRC6 protein induces the destabilization and translational repression of the target mRNA.

**Figure 2 ncrna-07-00057-f002:**
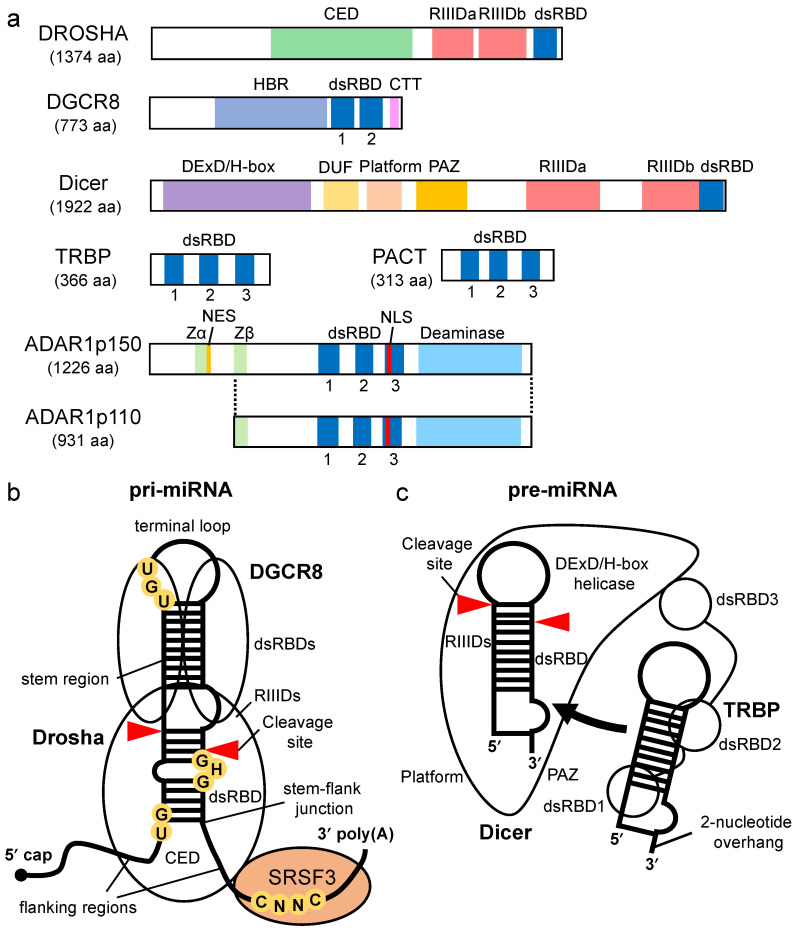
Domain structure of proteins involved in miRNA biogenesis and structural features of processing complexes of pri-miRNA and pre-miRNA. (**a**) The domain structures of Drosha, DGCR8, Dicer, TRBP, PACT, ADAR1p150, and ADAR1p110. CED indicates central domain; RIIIDa, RNase III domain a; RIIIDb, RNase III domain b; dsRBD, double-stranded RNA binding domain; HBR, heme-binding region; DExD/H-box, DExD/H-box helicase; CTT, C-terminal tail; DUF283, domain of unknown function; PAZ, Piwi–Argonaute–Zwille; Zα and Zβ, Z-DNA binding domain α and β, respectively; NES, nuclear export signal; NLS, nuclear localization signal. (**b**) Pri-miRNA has a terminal loop, stem region, and flanking region as common structural features, and some pri-miRNAs have conserved sequence motifs. UG is recognized by Drosha, UGU is recognized by DGCR8, GHG is recognized by dsRBD of Drosha, and CNNC is recognized by Ser/Arg-rich splicing factor 3 (SRSF3). (**c**) Pre-miRNA is structurally characterized by a terminal loop, a stem region, and a 3′ end overhang.

**Figure 3 ncrna-07-00057-f003:**
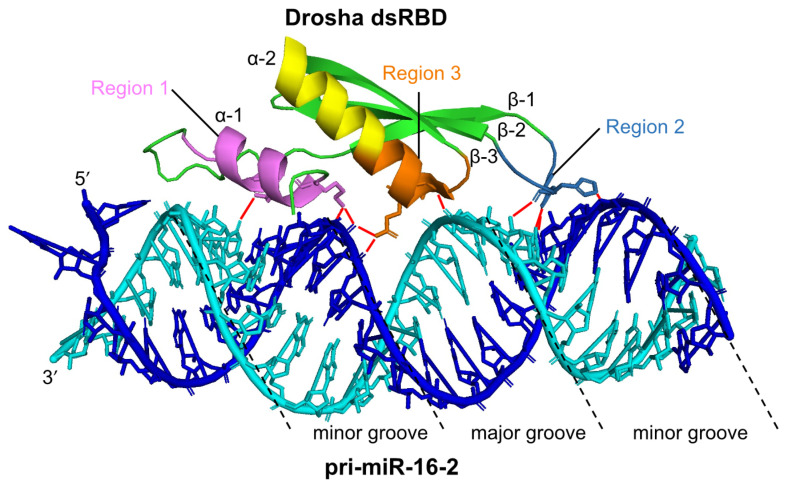
The structure of human Drosha dsRBD interacting with pri-miR-16-2. The dsRBD binds to the stem region of pri-miRNA via three regions (Region 1: violet, Region 2: sky-blue, Region 3: orange). A pri-miR-16-2 RNA strand is colored in blue, and the other RNA strand is colored in cyan. Red lines indicate the interaction sites between Drosha dsRBD and pri-miR-16-2. This figure is modified from the report produced by Partin et al. [52] (Protein Data Bank ID: 6V5B) using PyMOL (The PyMOL Molecular Graphics System, Version 2.0 Schrödinger, LLC).

**Figure 4 ncrna-07-00057-f004:**
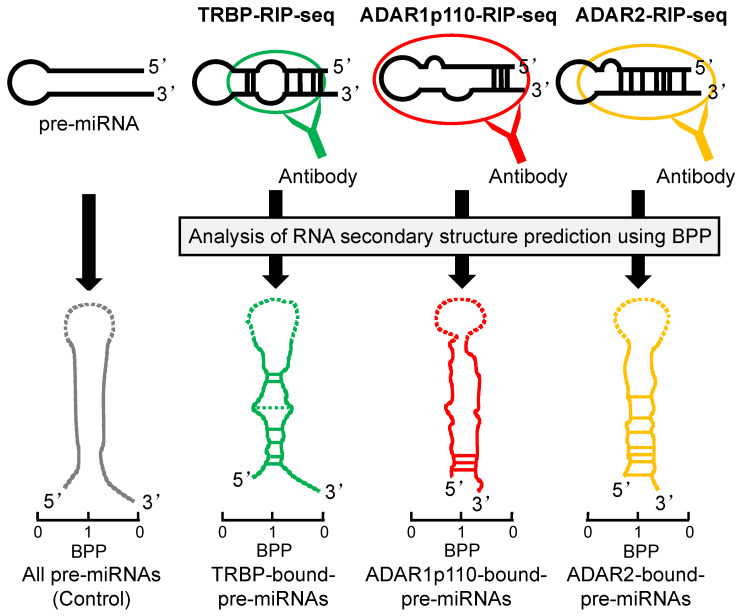
The pre-miRNA secondary structures preferentially bound by TRBP, ADAR1p110, or ADAR2. Secondary structures of pre-miRNAs, preferentially bound by TRBP, ADAR1p110, or ADAR2 as predicted by BPP. The pre-miRNAs bound to each dsRBP was determined by RIP-sequencing and secondary structures of pre-miRNAs are shown here schematically, modified from ref [93,97]. BPP is calculated for each base of pre-miRNAs and is a value in the range of 0 to 1; 0 indicates the low base-pairing probability and 1 indicates the high probability. The horizontal lines between 5′ strand and 3′ strand represent base-pairing, and the solid lines indicate the nucleotide positions with significantly high BPPs relative to the control and the dotted lines indicate the positions with significantly low BPPs. The BPP of the terminal loop of pre-miRNA was not calculated and shown in dotted line.

**Figure 5 ncrna-07-00057-f005:**
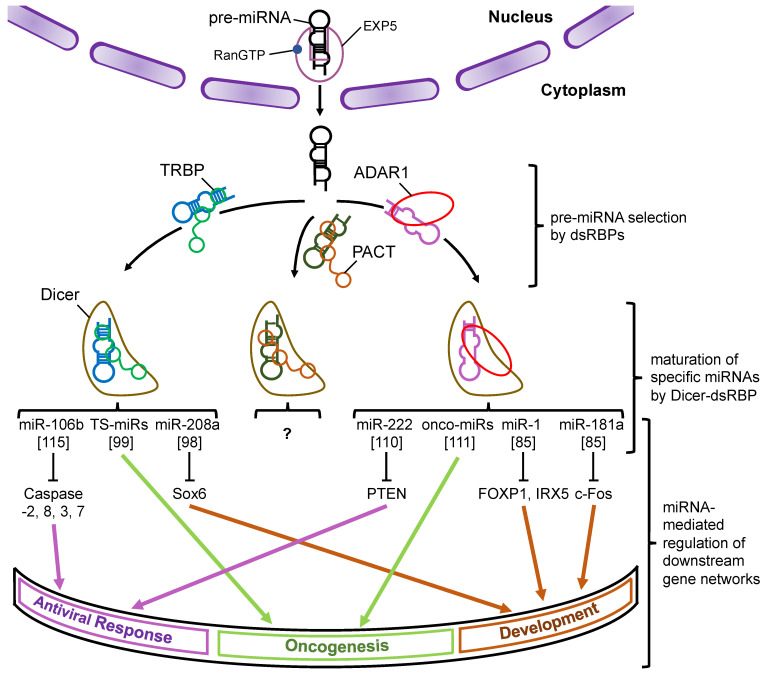
A schematic model of the miRNA–mRNA gene expression network through the modulation of Dicer-mediated processing by its associated proteins. Pre-miRNAs are transported from the nucleus to the cytoplasm by Exp5, which couples with RanGTP. Each dsRBP (TRBP, PACT, or ADAR1) binds to the pre-miRNAs based on their own substrate specificity and recruits them to Dicer to facilitate their maturation. The miRNAs shown in the figure represent the typical miRNAs that are specifically processed by Dicer–TRBP or Dicer–ADAR1 interaction and regulate the downstream gene expression networks. To date, the miRNAs specifically processed by Dicer–PACT interaction have not yet been identified. Note that the pathways shown here were reported by different studies. The numbers in the square brackets indicate the reference numbers.

## Data Availability

Not applicable.

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
