# Peer review of "Modulation of MicroRNA Processing by Dicer via Its Associated dsRNA Binding Proteins"

_ncrna, 2021, doi:10.3390/ncrna7030057_

Round 1
Reviewer 1 Report
In this review, Yoshida et al provide an overview of microRNA processing in higher eukaryotes, and present into this context their recent results of TRBP-bound structured pre-miRNAs (ref 90) as well as the interesting connection between miRNA processing and ADAR proteins, which is still not well understood and an active research topic. This short review is well-written and provides a clear primer into the field, and the figures are clear and well-presented (with one exception given below).
The review may benefit from citing a recent important paper in the field of miRNA biogenesis, which found that suboptimal hairpin pri-miRNAs can be assisted by proximal hairpin processing through Microprocessor multimerization (Fang & Bartel, 2021, Mol Cell). (either in section 2, or in the “processing enhancement” section 4)
Furthermore, the different substrate recognitions by different Dicers can be linked to the substrate termini (Sinha et al, 2018, Science), an important paper that may be cited in section 3 (pages 5/6).
The lower panel hairpins in Figure 4 are a bit unclear. Are the hairpin ‘sides’ drawn or plotting a measured value? The BPP scale (0,1,0) also would need to be clarified. In the present form the figure only seems to convey that TRBP and ADAR2 bind structured pre-miRNAs, while ADAR1p110 binds less structured pre-miRNAs.
For section 5.1, the authors may also choose to cite (Reich et al, 2018, Genes & Dev), showing that C. elegans ADAR antagonizes the miRNA pathway likely by substrate competition (along with refs 98,99).
In the concluding remarks, the authors propose that the second challenge would be more accurate RNA prediction algorithms to decipher miRNA structures. An alternative to this, that the authors could mention, would be the usage of next-generation sequencing techniques based on RNA structure, such as SHAPE-seq (Watters et al, 2016, Methods) or Structure-seq (Ding et al, 2014, Nature), and similar methods using distinct adducts/crosslinkers (EDC, glyoxal, etc.)
Some minor adjustments to the English of the text could clarify some sentences, such as for example:
page 1 line 9: “each miRNA is capable of recognizing and regulating many types of mRNAs simultaneously”, the word “simultaneously” should be removed, as these regulations are not simultaneous (i.e. one molecule of miRNA does not bind several differents mRNAs at the same time)
line 13, the authors may want to add just a small parenthesis to define ‘heterochronic’ genes (can be as simple as “mutants that undergo development/differentiation at an abnormal time within the organism”)
Reviewer 2 Report
First of all, I would like to highlight that the authors have done an excellent review of the modulation of microRNA processing mediated by double-stranded RNA-binding proteins. However, some additional considerations are needed:
- Please provide more detailed information on the structure of double-stranded RNA-binding domains (dsRBDs) and, in particular, on the importance of the length of the L1 region in Drosha and the consequences of expanding this region in other well-known dsRBDs, such as Dicer. This reference may be useful:
Zhang, Xiaoxiao, et al. "Extending the L1 region in canonical double-stranded RNA-binding domains impairs their functions." Acta Biochimica et Biophysica Sinica 53.4 (2021): 463-471.
- Please include a description of plant dicer-like proteins (especially DCL1) and other plant proteins with dsRBDs that are involved in microRNA processing. Where possible, compare with their animal counterpart. This reference may be useful:
Fukudome, Akihito, and Toshiyuki Fukuhara. "Plant dicer-like proteins: double-stranded RNA-cleaving enzymes for small RNA biogenesis." Journal of plant research 130.1 (2017): 33-44.
